**Data Availability Statement:** All relevant data are within the paper.

**Funding:** The experiment was supported from Special fund support project for clinical medicine

# Towards low cost, multiplex clinical genotyping: 4-fluorescent Kompetitive Allele-Specific PCR and its application on pharmacogenetics

**Wei Suo**📱**, Xiujin Shi, Sha Xu, Xiao Li, Yang Lin**\*

Pharmacy Department, Beijing Anzhen Hospital, Capital Medical University, Beijing, China

\* bjylnh@163.com

## Abstract

Single-nucleotide polymorphisms (SNPs) is associated with efficacy of specific drugs. Although there are several methods for SNP genotyping in clinical settings, alternative methods with lower cost, higher throughout and less complexity are still needed. In this study, we modified Kompetitive Allele Specific PCR to enable multiplex SNP genotyping by introducing additional fluorescent cassettes that specifically help to differentiate more amplification signals in a single reaction. This new format of assay achieved a limit of detection down to 310 copies/ reactions for simultaneous detection of 2 SNPs with only standard end-point PCR workflow for synthetic controls, and genotyped 117 clinical samples with results that were in 100% agreement with hospital reports. This study presented a simplified, cost-effective high-throughput SNP genotyping alternative for pharmacogenetic variants, and enabled easier access to pharmaceutical guidance when needed.

## Background

Single-nucleotide polymorphisms (SNPs) is associated with efficacy of specific drugs. For pharmacogenomic purpose SNP test is extensively used to guide clinical decisions. For example, Polymorphisms of the CYP2C9 (cytochrome P450, family 2, subfamily C, polypeptide 9) gene (CYP2C9*2, CYP2C9*3) and the VKORC1 (vitamin K epoxide reductase complex, subunit 1) gene ($-1639G > A$) can pose major impact on the maintenance dose of certain drugs such as Warfarin. Genotyping target SNPs for patients provides a time-saving method to guide proper maintenance dose, while reducing the adverse reaction risks and reoccurrence of thromboembolic episodes. [1] The genotyping of glucose-6-phosphate dehydrogenase (G6PD) is another typical example which demonstrates how population genotyping provides guidance in personalized medicine. G6PD deficiency prevalence is found to be high in malaria-endemic regions, while dominant antimalarial drugs can cause hemolysis to various degrees in G6PD deficient patients. Screening of G6PD deficiency alleles in high-risk populations would provide valuable estimation of hemolysis risk when using drugs such as primaquine, thereby improving public drug safety.[2]

development of Beijing Municipal Hospital Administration(ZYLX201805) and Major new drug creation national major science and technology special project "construction of clinical research demonstration platform for cardiovascular disease drugs"(2017ZX09304017). The funders had no role in study design, data collection and analysis, decision to publish, or preparation of the manuscript.

**Competing interests:** The authors have declared that no competing interests exist.

Many methods exist for clinical SNP genotyping. Majority of methods that can be used in clinical settings are PCR-based [3], which include, probe based qPCR assay that detect SNP by allele specific probes labeled with difference fluorescent dye [4]; allele-specific PCR (ASPCR) using SNP-containing primers only direct amplification on their complementary allele [5]; melting curve analysis (high-resolution melting analysis) utilizing saturated intercalating dye that would display the change of melting curve caused by slight difference of Tm between two SNP alleles [6]; and Cold-PCR which amplifies minority alleles selectively from mixtures of wild-type, utilizing the difference of dsDNA-dissociation by precisely controlled denaturing temperature [7]. These chemistries represent the most commercial kit now used for clinical diagnosis of SNP. Other methods also exist, such as Mass Array [2], Capillary electrophoresis [8] and digital PCR [9]. Although being high-throughput or super sensitive, these methods highly rely on the post-PCR analysis, which is time-consuming, vulnerable to cross contamination and equipment-demanding, therefore hindering the application in clinical settings where turnaround time, robustness and simplicity is crucial.

Improved from ASPCR, Kompetitive Allele Specific PCR (KASP, LGC Group) introduced fluorescence resonance energy transfer (FRET) for signal generation, where 2 fluorescent cassettes are used for identification of allele-specific amplification for a single bi-allelic SNP. Compared with other alternatives, KASP is more flexible [10], cost-effective [11] and robust [12], and therefore is now widely applied in large scale genotyping for many areas including agriculture [13] and human health [14, 15]. However, due to KASP's closed platform, the multiplexity is limited to only single bi-allelic SNP in each reaction, restricting the further improvement of the throughput and efficiency.

In Beijing Anzhen Hospital, we provided 22,593 patients with genotype test and pharmaceutical guidance report service in 2018. Average number of SNPs tested for each patient is between 2~5. The current kit adopted for these tests utilize a ligase chain reaction-based technology that detect single SNP in each reaction, which means at least over 22,000 separate reactions are needed for the above tests. If 2 or even more SNPs can be tested in 1 reaction, more than 10, 000 reactions can be saved, enabling lower cost and reduced turnaround time. To address this problem, we aim to develop an enhanced version of KASP: 4-fluorescent KASP, which simultaneously utilize more fluorescent channels than just FAM/ Hex, and thereby enabling multiplexing genotyping. Here we describe the feasibility of genotyping with the new cassette with Cal Red 610 and Quasar 670 labels, and the compatibility of the new cassette with those contained in standard KASP reagent.

## Method

### Design of fluorescent cassette

Following FRET principle, one fluorescent dye and one BHQ-2 quencher were attached to 5' and 3' ends of two reverse complementary single stranded DNA separately. The sequence was blasted to confirm that it would not amplify any natural sequences. These sequences were sent to LGC, Biosearch Technologies for synthesis. Multiple pairs of cassettes were designed for empirical selection. The final selected sequences are as follows,

### Design of KASP primers

The allele specific primer was designed by following the principles described previously[5], and validated with Primer-BLAST tool against NCBI database. The sequence of the used primers is available upon request. For standard KASP, sequences of specific primers were respectively added at 5' end by FAM tag or Hex tag following previous published papers [16]. For modified KASP which involved fluorescent other than FAM/ Hex, and tails for specific

fluorescent cassette were added, which were CAL Fluor Red 610 tag or Quasar 670 tag. Oligos were ordered through LGC, Biosearch Technologies. The sequences are as follows,

## Design of positive control

Positive control for each target was designed and ordered through LGC, Biosearch Technologies. The sequence was designed by including primer binding site for allele specific sequences of each primer, with 'tt' as spacer. The sequences are as follows,

## Clinical samples

This study was approved by the Clinical Research Ethics Committee of Beijing Anzhen Hospital. As the samples were collected for clinical tests following routine protocol and we used only small portion of the retention before it was discarded, the Clinical Research Ethics Committee of Beijing Anzhen Hospital agree to exempt the informed consent. The clinical DNA, extracted from blood samples between Oct. 12th to Jan. 6th 2020, was provided with clinically diagnosed genotype. The information was accompanied with sample ID only, personal information were not revealed for privacy purpose.

**Clinical diagnostics.** Clinical diagnostic in our hospital was done using a commercial kit from Sino Era Genotech (http://www.sino-era.com/, Beijing, China). The kit includes reagents for sample preparation, molecular beacon probes and hybridization buffer. Briefly, 200 μL of whole blood was added to 1 mL of red blood cell lysis buffer and incubated for 5 minutes at room temperature. The mixture was centrifuged (3000 rpm) for 5 minutes and the supernatant removed. 50 μL preservation solution was added to resuspend the precipitate. 1.5 μL of sample was then combined with detecting reagent and analyzed with the TL998A real-time PCR system (Tianlong, Xi'an, China). The detecting reagent includes hybridization buffer and two molecular beacon probes labeled with FAM and HEX for wild-type and mutation, respectively.

## PCR protocol and data analysis

For standard KASP, the reaction was done with KASP V4.0 2X Master Mix (Low Rox, LGC Biosearch Technologies) following manufacturer's instructions. For assays with additional fluorescent channels, 0.08 μM of both fluorescent oligos and 0.32 μM of corresponding quencher oligos were added to BHQ probe mastermix (LGC Biosearch Technologies), or KASP V4.0 2X Master Mix. For each SNP, 0.15 μM of each allele specific primer and 0.4 μM of common reverse primer were used. The reaction volume in this study was 10 μl unless other wisely specified. The PCR protocol was the same as standard KASP (36 cycles of amplification in total), except that all fluorescent channel was read at the final step. The reaction was done with Biorad CFX96 and analyzed by Bio-Rad CFX Maestro 1.1 (Version:4.1.2433.1219) software.

## Result

### Evaluation of cassettes with Cal Red 610 and Quasar 670

Feasibility of doing self-made KASP with fluorescent channels other than FAM/ Hex was evaluated by adding newly designed fluorescent cassette labeled with Cal Red 610 and Quasar 670 to routinize PCR master mix (BHQ probe Mastermix, LGC Biosearch Technologies). Synthetic controls for rs1057910 and rs4986893 were assayed with this self-made KASP system, and all 16 controls were correctly clustered for both assays (Fig 1), indicating that this new pair of

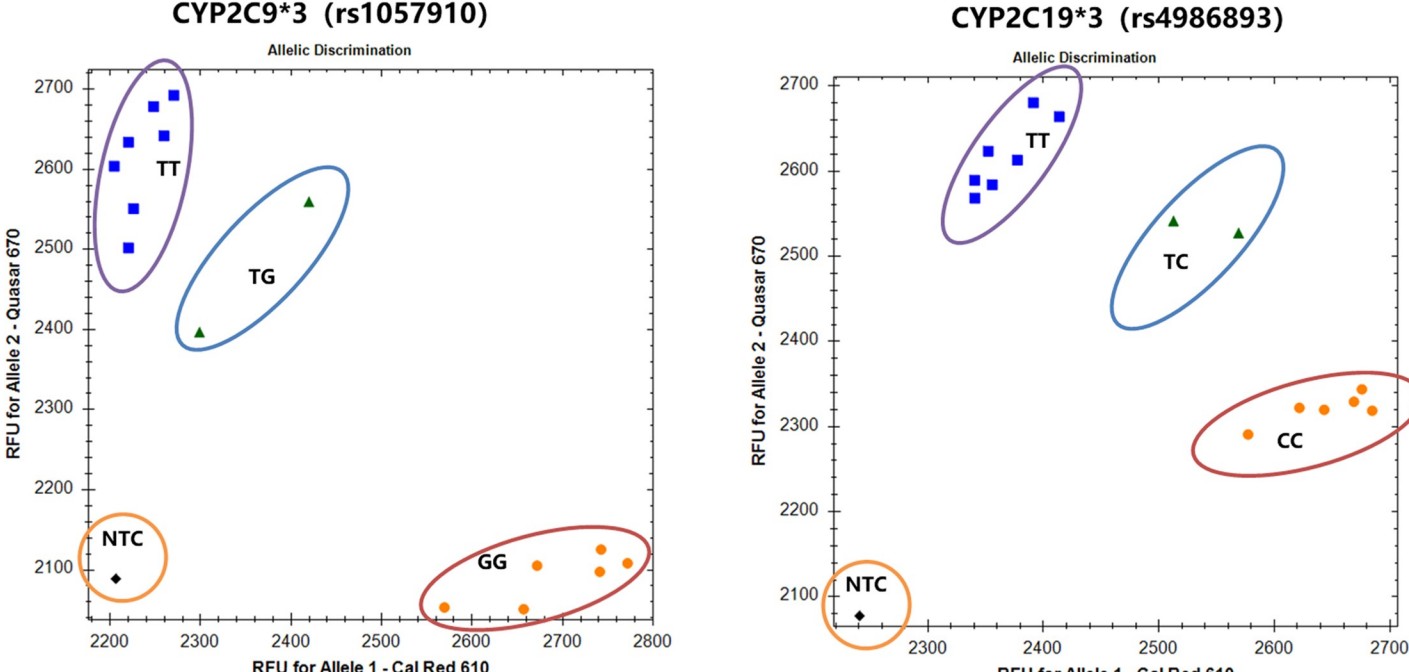

**Fig 1. Genotyping of rs1057910 (left) and rs4986893 using cassette with Cal Red 610 and Quasar 670 in routine qPCR master mix following KASP protocol.**
Allele specific primers were added with tail for Cal Red 610 (G for rs1057910, and C for rs4986893) or Quasar 670 (T for rs1057910, and T for rs4986893). Synthetic positive control of each gene was genotyped individually (homozygotes being approximately 300,000 copies/ reactions, and heterozygotes being 150,000 copies of each allele per reaction). One None-Template Control (NTC) was included in each run.

cassettes with Cal Red 610 and Quasar 670 works fine both for rs1057910 and rs4986893 in routine qPCR master mix.

## Evaluation of new cassette in standard KASP reagent

To confirm the compatibility of the new cassette with standard KASP chemistry, cassettes with Cal Red 610 and Quasar 670 were added to standard KASP, and genotyped clinical samples as well as synthetic controls for rs1057910 and rs4986893. The controls are all correctly genotyped, and the result for the clinical samples is in 100% agreement with that from hospital report (Table 1), indicating that the newly designed cassettes with Cal Red 610 and Quasar 670 work well within standard KASP reagent.

## Limit of detection of 4-fluorescent KASP

The limit of detection (LOD) was determined as the minimal amount of positive control added to each reaction that produced a correct clustering. The LOD was determined to be 310 copies/ reactions, or 5 μl of 62 copies/ μl positive control (Fig 2), indicating that this new

| Sequence Name | 5' modification | sequence | 3' modification |
|---|---|---|---|
| Cal | CAL Fluor Red 610 | TAG AAG GCA CAG TCG AGG | |
| Cal quencher | | CGA CTG TGC CTT CTA | BHQ-2 |
| Qua | Quasar 670 | GTA AAA CGA CGG CCA GTG | |
| Qua quencher | | GGC CGT CGT TTT AC | BHQ-2 |

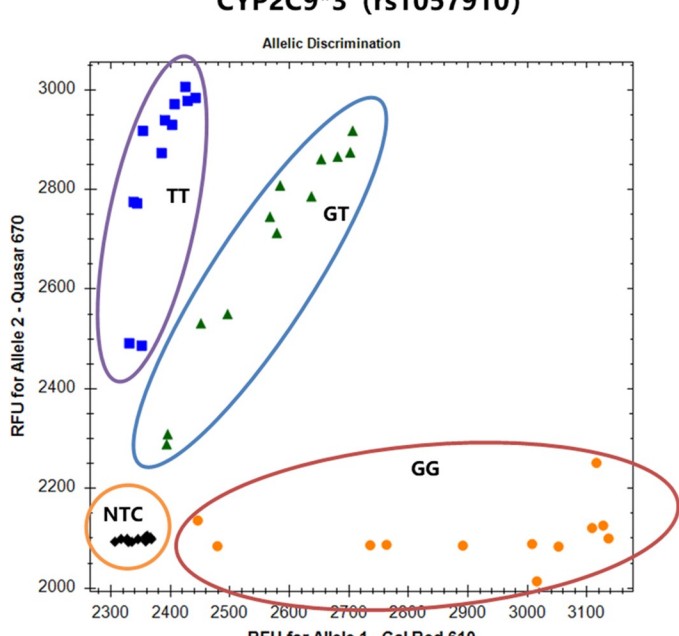

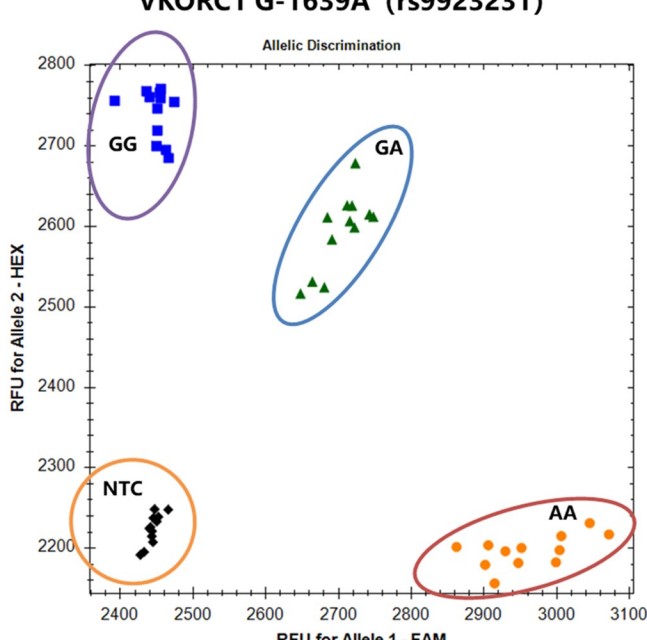

**Fig 2. Limit of detection of 4-fluorescent KASP.** Allele specific primers were added with tails for Cal Red 610 (G for rs1057910), Quasar 670 (T for rs1057910), FAM (A for rs9923231) and Hex (G for rs9923231). Positive control of rs1057910 and rs9923231 were diluted by 10-fold from 6.23 x $10^{10}$ copies/ µl (100 nM stock solution) to 6.2 copies/ µl. The two synthetic genes were diluted together to make duplex-SNP controls, where allele-G for rs1057910 and allele-A for rs9923231 were put in one solution and the other two alleles were put in another solution. The heterozygote controls were made by mixing homozygote solution of both alleles with 1:1 ratio at each concentration. 5 µl of each concentration was used in each reaction. For each concentration, each genotype was run in duplicates, with a total of 12 NTCs included. The data above showed the concentrations from 62 copies/ µl to 6.23 x $10^{6}$ copies/ µl (6 concentrations in total). The 6.2 copies/ µl samples were not clustered correctly and therefore not included.

4-fluorescent KASP was highly sensitive in all four channels, and could meet the needs of clinical detection.

## Discussion

Clinical SNP test is critical for prescribing specific drugs [1, 2]. In Beijing Anzhen Hospital, we provided 22,593 patients with genotype test and pharmaceutical guidance report service in 2018. The currently used genotyping assays were approved by the State Drug Administration of China, all of which involved more than just 1 SNP. These assays can only detect 1 SNP in each reaction, therefore multiple reactions are required for each patient. In addition, the cost is up to $ 60 per SNP, limiting the access of the low-income patients and thus exposing them to higher risks. Alternative method with higher throughout and lower cost is imperatively needed.

Although Mass-array and NGS offer super high multiplexibility, ranging from tens to even thousands of SNPs with reasonable price, they do not fit clinical needs at Anzhen hospital. As only 2~5 SNPs are required for each patient, and the result shall be returned within 24 hours, rendering both Mass-array and NGS incompatible in this scenario. To address this problem, we also considbuquered probe-based multiplex qPCR, which efficiently detects 2~3 SNPs in one reaction, but the perplexity of designing both primers and probes for each SNP, and the burden of fine tune PCR conditions for each assay is challenging.

With this new 4-fluorescent KASP assay available, the output for SNP genotyping will be doubled without requiring additional labor or significant cost. The workflow is exactly the

| Sequence name | Sequence |
|---|---|
| rs1057910-FAM | GCGATTAGCCGTTAGGATGAGGTGGGGAGAAGGTCAAT |
| rs1057910-HEX | GTCGGTGAACAGGTTAGAGA GCTGGTGGGGAGAAGGTCAAG |
| rs1057910-Quasar 670 | GTA AAA CGA CGG CCA GTGGCTGGTGGGGAGAAGGTCAAG |
| rs1057910-Cal Red 610 | TAG AAG GCA CAG TCG AGGGGTGGGGAGAAGGTCAAT |
| rs1057910-Common | GGAGCCACATGCCCTACACAGAT |
| rs4986893-FAM | GCGATTAGCCGTTAGGATGAAAC TTG GCC TTA CCT GGA TC |
| rs4986893-HEX | GTCGGTGAACAGGTTAGAGA AAC TTG GCC TTA CCT GGA TT |
| rs4986893-Quasar 670 | GTA AAA CGA CGG CCA GTGAAC TTG GCC TTA CCT GGA TT |
| rs4986893-Cal Red 610 | TAG AAG GCA CAG TCG AGGAAC TTG GCC TTA CCT GGA TC |
| rs4986893-Common | AAACATCAGGATTGTAAGCAC |
| rs9923231-FAM | GCGATTAGCCGTTAGGATGATTG AGC CAC CGC ACC A |
| rs9923231-HEX | GTCGGTGAACAGGTTAGAGA TAG CCA CCG CAC CG |
| rs9923231-Quasar 670 | GTA AAA CGA CGG CCA GTGTAG CCA CCG CAC CG |
| rs9923231-Cal Red 610 | TAG AAG GCA CAG TCG AGGTTG AGC CAC CGC ACC A |
| rs9923231-Common | GGAAGTCAAGCAAGAGAAGACCTGAA |

same as standard KASP, which is already well known for its simplicity and flexibly [10, 12], and the additional cassette added to the assay is only single-labeled oligos, which is significantly less expensive as compared with dual-labeled oligos used for routine multiplex genotyping. The cost can be further reduced by making this single-labeled oligos in bulk; these single-labeled oligos are not gene-specific, and therefore applicable for genotyping of all SNPs. The universal-nature of these oligos bring in at least two benefit: 1, the unit cost (per nmol) of synthesizing oligos goes down significantly as the scale goes up. The universal oligos that can be made in one large batch for all SNPs will be more cost-effective than those individually customized, small scale probes; 2, the perplexity of designing both primers and probes for each SNP no longer exists, as only gene specific primers are needed, leading to improved convenience and flexibility for designing new assays for multiplexing SNP analysis.

The limitations of current study include limited scale of clinical validation, and small number of SNP sites evaluated for dual-SNP assays. To further support 4-fluorescent KASP's capacity on clinical genotyping, experiments that involve more clinical samples, and various SNP sites should be conducted.

Current result, to our knowledge, is the first demonstration of performing KASP for multiple SNPs in one reaction. Although we only used 2 pairs of cassettes with 4-fluorescent channels for simultaneously genotyping of 2 SNPs in one reaction, there is the potential that more

| Sequence name | Sequence |
|---|---|
| rs1057910control-G | GCTGGTGGGGAGAAGGTCAAGttATCTGTGTAGGGCATGTGGCTCC |
| rs1057910control-T | GGTGGGGAGAAGGTCAATttATCTGTGTAGGGCATGTGGCTCC |
| rs4244285control-A | TTCCCACTATCATTGATTATTTCCCAttAAGAACACCAAGAATCGATGGA |
| rs4244285control-G | TTCCCACTATCATTGATTATTTCCCGttAAGAACACCAAGAATCGATGGA |
| rs4986893control-C | AAC TTG GCC TTA CCT GGA TCttGTGCTTACAATCCTGATGTTT |
| rs4986893control-T | AAC TTG GCC TTA CCT GGA TTttGTGCTTACAATCCTGATGTTT |
| rs9923231control-A | TTG AGC CAC CGC ACC AttTTCAGGTCTTCTCTTGCTTGACTTCC |
| rs9923231control-G | TAG CCA CCG CAC CGttTTCAGGTCTTCTCTTGCTTGACTTCC |

**Table 1. Genotyping result reported by hospital and by KASP using new cassettes.**

| Method | Genotyping results (n = 117) | | | | | |
|---|---|---|---|---|---|---|
| | rs1057910 (n = 32) | | | rs4986893 (n = 85) | | |
| | G | GT | TT | C | CT | TT |
| Genotype reported by hospital | 30 | 2 | 0 | 78 | 6 | 1 |
| KASP using new cassettes | 30 | 2 | 0 | 78 | 6 | 1 |

cassettes can be added and thus more SNPs and even multi-allelic SNPs can be evaluated in a single reaction, considering that 6 or more channels plate reader and qPCR machines are emerging.

# Author Contributions

**Conceptualization:** Wei Suo, Yang Lin.

**Data curation:** Wei Suo, Xiao Li.

**Formal analysis:** Wei Suo.

**Funding acquisition:** Wei Suo.

**Investigation:** Wei Suo.

**Methodology:** Wei Suo.

**Project administration:** Wei Suo, Yang Lin.

**Resources:** Wei Suo, Yang Lin.

**Software:** Wei Suo.

**Supervision:** Wei Suo, Yang Lin.

**Validation:** Wei Suo.

**Visualization:** Wei Suo.

**Writing – original draft:** Wei Suo.

**Writing – review & editing:** Wei Suo, Xiujin Shi, Sha Xu, Yang Lin.

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
