## [Decision Letter · Decision Letter 0]

2 Jan 2020

PONE-D-19-32764

Towards low cost, multiplex clinical genotyping: 4 fluorescent Kompetitive Allele-Specific PCR and its application on pharmacogenetics

PLOS ONE

Dear Mrs. Lin,

Thank you for submitting your manuscript to PLOS ONE. After careful consideration, we feel that it has merit but does not fully meet PLOS ONE’s publication criteria as it currently stands. Therefore, we invite you to submit a revised version of the manuscript that addresses the points raised during the review process.

The manuscript should be prepared in accordance with the rules of the journal. All information and details should be fully presented in the work.

The English style must be checked by a native English speaker or a specialized language service.

We would appreciate receiving your revised manuscript by Feb 16 2020 11:59PM. To enhance the reproducibility of your results, we recommend that if applicable you deposit your laboratory protocols in protocols.io, where a protocol can be assigned its own identifier (DOI) such that it can be cited independently in the future. For instructions see: http://journals.plos.org/plosone/s/submission-guidelines#loc-laboratory-protocols

We look forward to receiving your revised manuscript.

Kind regards,

Ruslan Kalendar, PhD

Academic Editor

PLOS ONE

Journal Requirements:

3. We noticed you have some minor occurrence(s) of overlapping text with the following previous publication(s), which needs to be addressed:

https://doi.org/10.1373/clinchem.2008.115295

https://doi.org/10.1016/j.jmoldx.2017.05.007

In your revision ensure you cite all your sources (including your own works), and quote or rephrase any duplicated text outside the Methods section. Further consideration is dependent on these concerns being addressed.

4. Please note that all PLOS journals ask authors to adhere to our policies for sharing of data and materials: https://journals.plos.org/plosone/s/data-availability. According to PLOS ONE’s Data Availability policy, we require that the minimal dataset underlying results reported in the submission must be made immediately and freely available at the time of publication. As such, please consider including the minimal information (fluorescent cassette sequence and primer sequences) upon publication.

5. Please provide additional details regarding participant consent. In the ethics statement in the Methods and online submission information, please ensure that you have specified (1) whether consent was suitably informed and (2) what type you obtained (for instance, written or verbal). If your study included minors under age 18, state whether you obtained consent from parents or guardians. If the need for consent was waived by the ethics committee, please include this information.

Reviewers' comments:

Reviewer's Responses to Questions

**Comments to the Author**

1. Is the manuscript technically sound, and do the data support the conclusions?

Reviewer #1: Yes

Reviewer #2: Partly

2. Has the statistical analysis been performed appropriately and rigorously? 

Reviewer #1: No

Reviewer #2: No

3. Have the authors made all data underlying the findings in their manuscript fully available?

Reviewer #1: No

Reviewer #2: Yes

4. Is the manuscript presented in an intelligible fashion and written in standard English?

Reviewer #1: No

Reviewer #2: No

5. Review Comments to the Author

Reviewer #1:

This paper describes an incremental advance that is unlikely to be adopted because there are many better and more accessible methods for genotyping. While the work demonstrates multiplexed genotyping, the level of multiplexing is very limited.

Specific comments:

Too few samples with rare genotypes are tested. It is important to show that there are no errors with rare genotypes.

The statement in the materials and methods "Sequence of used primers is available upon request" is not acceptable for PLOS One. Authors should include all sequence information in the paper or Supplementary Information section. It is not optional information that should have to be requested.

English language needs to be reviewed by a native English speaker.

Reviewer #2:

In the current study, a modified Kompetitive Allele Specific PCR, in which 4 fluorescents were used, was described. Further, this modified KASP was used and evaluated. Overall, this job is an improvement for the routine KASP that can be a cost-effective high-throughput SNP genotyping alternative for pharmacogene variants. There are many mistakes in the text concerning English wording and grammar. It is suggested to authors to co-opt a native English speaker when revising the manuscript.

Specific Comments

1. Why did rs1057910 and rs4986893 were selected to evaluated the modified KASP?

2. the result of modified KASP for clinical samples was in 100% agreement with that from hospital report. The authors should describe what kind of method the hospital reports based on. Moreover, the number of the samples (n=40) that were carried out to evaluate the mothed was extremely limited. The authors should increase sample size.

3. Three-line table should be used in the manuscript.

4. The format of reference in the text doesn’t meet the standard of PLOS ONE.

6. PLOS authors have the option to publish the peer review history of their article (what does this mean?). If published, this will include your full peer review and any attached files.

Reviewer #1: No

Reviewer #2: No

---

## [Author Response · Author response to Decision Letter 0]

16 Feb 2020

Dear Editor,

Thank you for the kind suggestions. We have updated the manuscript accordingly by addressing to all points raised both you and the reviewers.

For Journal Requirements:

1. we have updated our manuscript to meet PLOS ONE's style requirements, including those for file naming. 

2. we have no supporting info for this manuscript, all relevant info is now incorporated in main body of the manuscript.

3. we have revised the wording and cited all sources, which is now in line 35 to 48. 

4. we have listed fluorescent cassette sequence and primer sequences in revised manuscript, which is now in line 92, line 102 and line 107.

5. we have updated ethical statement in line 110 : “This study was approved by the Clinical Research Ethics Committee of Beijing Anzhen Hospital. As the samples were collected for clinical tests following routine protocol and we used only small portion of the retention before it was discarded, the Clinical Research Ethics Committee of Beijing Anzhen Hospital agree to exempt the informed consent. “

For Reviewers' comments:

Reviewer #1:

1, Too few samples with rare genotypes are tested. It is important to show that there are no errors with rare genotypes.

Response: due to limited number of available clinical samples, the sample size in previous version of manuscript is indeed small. We have now enrolled more participants, and evaluated our assay in 117 clinical samples. The manuscript has been updated accordingly. It is indeed important to evaluate new assays with as many genotypes as possible, including those rare genotypes. Current study, however, focuses only on germline mutation that is currently clinically relevant in our hospitals, we do not yet have resource to explore more rare genotypes. But we would surely consider cooperate with other organizations that have such need and resources in future. 

2, The statement in the materials and methods "Sequence of used primers is available upon request" is not acceptable for PLOS One. Authors should include all sequence information in the paper or Supplementary Information section. It is not optional information that should have to be requested.

Response: we have now disclosed final version of all sequences in the manuscript, which is now in line 92, line 102 and line 107.

3, English language needs to be reviewed by a native English speaker.

Response: we have the manuscript reviewed by a native English speaker.

Reviewer #2:

1. Why did rs1057910 and rs4986893 were selected to evaluated the modified KASP?

Response: Current study focuses on SNP that is currently clinically relevant in our hospitals. rs1057910 and rs4986893 is among the most common sites that is being evaluated daily. Although there is a couple of other sites, we are not yet able to completed all of them due to limited resources.

2. the result of modified KASP for clinical samples was in 100% agreement with that from hospital report. The authors should describe what kind of method the hospital reports based on. Moreover, the number of the samples (n=40) that were carried out to evaluate the mothed was extremely limited. The authors should increase sample size.

Response: our hospital uses a commercial kit from Sino Era Genotech (http://www.sino-era.com/, Beijing, China). Detailed info and protocol has been added to revised manuscript in line 119 to 127. We have also increase sample size to 117.

3. Three-line table should be used in the manuscript.

Response: all tables is now updated using a three-line format.

4. The format of reference in the text doesn’t meet the standard of PLOS ONE.

Response: format is now updated in accordance with PLOS ONE guideline.

---

## [Decision Letter · Decision Letter 1]

2 Mar 2020

Towards low cost, multiplex clinical genotyping: 4-fluorescent Kompetitive Allele-Specific PCR and its application on pharmacogenetics

PONE-D-19-32764R1

Dear Dr. Lin,

We are pleased to inform you that your manuscript has been judged scientifically suitable for publication and will be formally accepted for publication once it complies with all outstanding technical requirements.

With kind regards,

Ruslan Kalendar, PhD

Academic Editor

PLOS ONE

---

## [Editor Report · Acceptance letter]

4 Mar 2020

PONE-D-19-32764R1 

Towards low cost, multiplex clinical genotyping: 4-fluorescent Kompetitive Allele-Specific PCR and its application on pharmacogenetics 

Dear Dr. Lin:

I am pleased to inform you that your manuscript has been deemed suitable for publication in PLOS ONE. Congratulations! Your manuscript is now with our production department. 

With kind regards,

on behalf of

Dr. Ruslan Kalendar 

Academic Editor

PLOS ONE